# Field Screening of Wheat Advanced Lines for Salinity Tolerance

**Ehab S. A. Moustafa** [1], **Mohamed M. A. Ali** [2], **Mohamed M. Kamara** [3], **Mohamed F. Awad** [4], **Abdallah A. Hassanin** [5] **and Elsayed Mansour** [2,*]

1  Desert Research Center, Genetic Resources Department, Cairo 11753, Egypt; ehab.soudi@yahoo.com
2  Department of Crop Science, Faculty of Agriculture, Zagazig University, Zagazig 44511, Egypt; abd_lhamed@yahoo.com
3  Department of Agronomy, Faculty of Agriculture, Kafrelsheikh University, Kafr El-Sheikh 33516, Egypt; mohamed.kamara@agr.kfs.edu.eg
4  Department of Biology, College of Science, Taif University, P.O. Box 11099, Taif 21944, Saudi Arabia; m.fadl@tu.edu.sa
5  Genetics Department, Faculty of Agriculture, Zagazig University, Zagazig 44511, Egypt; dr.abdallah4@gmail.com
*  Correspondence: sayed_mansour_84@yahoo.es

**Abstract:** Salinity in soil or irrigation water requires developing genetically salt-tolerant genotypes, especially in arid regions. Developing salt-tolerant and high-yielding wheat genotypes has become more urgent in particular with continuing global population growth and abrupt climate changes. The current study aimed at investigating the genetic variability of new breeding lines in three advanced generations F6–F8 under salinity stress. The evaluated advanced lines were derived through accurate pedigree selection under actual saline field conditions (7.74 dS/m) and using saline water in irrigation (8.35 dS/m). Ninety-four F6 lines were evaluated in 2017–2018 and reduced by selection to thirty-seven F7 lines in 2018–2019 and afterward to thirty-four F8 lines in 2019–2020 based on grain yield and related traits compared with adopted check cultivars. Significant genetic variability was detected for all evaluated agronomic traits across generations in the salt-stressed field. The elite F8 breeding lines displayed higher performance than the adopted check cultivars. These lines were classified based on yield index into four groups using hierarchical clustering ranging from highly salt-tolerant to slightly salt-tolerant genotypes, which efficiently enhance the narrow genetic pool of salt-tolerance. The detected response to selection and high to intermediate broad-sense heritability for measured traits displayed their potentiality to be utilized through advanced generations under salinity stress for identifying salt-tolerant breeding lines.

**Keywords:** yield-related traits; genotypic and phenotypic coefficient of variation; broad-sense heritability; response to selection; genetic gain; cluster analysis; principal component analysis

## 1. Introduction

Wheat (*Triticum aestivum* L.) is a widespread staple food crop worldwide. It is a major source of energy and starch, as well as provides considerable amounts of dietary fiber, protein, and vitamins for human nutrition [1,2]. The total production amount of wheat in 2018 was 734.1 million tons, harvested from 214.3 million hectares [3]. Current and projected future population growth requires improving wheat production in response to worsening challenges due to climate change [4–6].

Salinity is one of the harsh environmental factors that devastatingly impact global wheat production [7,8]. Large areas of cultivated land around the world are salt-affected, particularly in arid regions due to low precipitation, high evaporation, poor drainage, poor irrigation practices, using saline water in irrigation, or rising water tables [7,9]. Under salinity conditions, wheat plants suffer from high osmotic stress, difficulties in nutrient uptake, and ion toxicity, which is reflected in reducing cell turgor and limiting growth and productivity [8,10].

The destructive impacts of salinity on wheat production can be mitigated by certain agronomic practices as utilizing a large amount of gypsum or by leaching to reduce salt from the soil, but these processes are expensive. On the other hand, growing salt-tolerant genotypes provides a long-term inexpensive practical solution and can be utilized in a large area which is the most effective approach [11]. Consequently, developing salt-tolerant and high-yielding wheat genotypes has become more urgent, especially with continuing global population growth and current climate changes [8]. However, improving salt-tolerant genotypes faces restricting factors as a lack of genetic diversity, limited selection effectiveness under salinity stress, and inadequate knowledge of complex salt-tolerance mechanism [9,10,12]. For these reasons, enhancing breeding for salt-tolerance is a valuable global concern to cope with the constraints on agricultural production [8].

Salt-tolerance can be characterized as the ability of the genotype to complete its growth cycle and produce appropriate grain yield under salinity stress compared to salt-sensitive genotypes [12]. Wheat genotypes possess diverse efficiency in producing acceptable grain yield under salinity stress [8]. Therefore, it is vital to address wheat genotypes under salinity conditions to identify salt-tolerant and sensitive ones. Evaluation of wheat genotypes under natural salinization is a crucial assessment approach, since the plants are screened under realistic and natural conditions such as soil heterogeneity, drought stress, and fluctuations of air temperature at the same time with salinity stress [7,13,14]. Thus, genotypic evaluation under natural saline field conditions assists in identifying suitable genotypes that could be grown under salinity conditions, as well as potential parents that could be integrated into breeding programs for salt-tolerance [8].

The objectives of this study were to: (i) investigate the genetic variability of breeding lines in three advanced generations, F6–F8, under salinity stress; (ii) identify superior advanced lines under salinity stress that can be exploited in the breeding program to develop further adapted cultivars for salt-affected regions; and (iii) estimate heritability, genetic gain and response to selection based on earliness and yield-related traits to determine their efficiency for selection in wheat under salinity stress.

## 2. Materials and Methods

### 2.1. Plant Material and Experimental Design

The data of this study were collected from three advanced generations, F6–F8, in a breeding program for salt-tolerance followed a pedigree scheme. The evaluated breeding lines were derived by crossing ten genotypes representing commercially adopted cultivars and exotic genotypes (Table S1). Ninety-four F6 lines were evaluated in 2017–2018 and reduced by selection to thirty-seven F7 lines in 2018–2019 and afterward to thirty-four F8 lines in 2019–2020. Grain yield and its components were the main selection criteria in the advanced generations compared with five adopted check cultivars (Table S1). The experimental design was alpha-lattice design with three replications. In the three generations, the plots consisted of six rows of 3 m long with a 20 cm spacing between rows.

### 2.2. Experimental Site and Agronomic Practices

Field experiments were performed during three growing seasons (2017–2018 to 2019–2020) at the Ras-Sudr Experimental Station, Desert Research Center, Southwest Sinai, Egypt (29.6° N and 32.7° E). The experimental site is arid and characterized by very low precipitation (accumulative annual precipitation is around 50 mm). The average monthly temperature and rainfall during the three growing seasons as well as the long-term average of 38 years are shown in Table S2. The experimental site is affected by salinity in both irrigation water (8.35 dS/m) and soil (7.74 dS/m). Chemical properties of the soil (at the depth of 0–30 cm) and irrigated water were analyzed and are presented in Table S3. Based on soil analysis, the soil is sandy loam throughout the profile (86.95% sand, 8.75% silt and 4.30% clay). The sowing dates in three generations were in the third week of November within the optimal period for wheat cultivation in the area. Phosphorus was applied before sowing at a rate of 31 kg P/ha as superphosphate (15.5% $P_2O_5$) and potassium was applied

at a rate of 80 kg K/ha as potassium sulfate (48% $K_2SO_4$). Nitrogen fertilizer was applied with a rate of 180 kg N/ha as ammonium sulfate (21% N) in five splits at 10-day intervals after sowing. There was no pest pressure in the two growing seasons. The used irrigation in this region for wheat is surface irrigation once every week.

### 2.3. Measured Traits

The measured traits were days to heading, plant height, number of grains per spike, 1000-grain weight, grain yield and biological yield. Days to heading was recorded as the number of days from sowing to the date when about 2 cm of awns were visible on 50% of stems in each plot. Plant height was measured as the distance in centimeters from the soil surface to the end of the spike. The number of grains/spike was recorded from ten randomly chosen spikes at each plot. Both grain yield and biological yield were recorded by hand-harvesting the four central rows from each plot and converting the weight to kilograms per hectare, based on the plot area that was harvested. The 1000-grain weight was determined as the weight of 1000 grains sampled from the harvest of four central rows.

### 2.4. Statistical Analysis

Data of the three generations were subjected to analysis of variance. The phenotypic, genotypic coefficients of variation were calculated according to the method suggested by Burton and Devane [15].

Broad-sense heritability was calculated as $h_b{}^2 = \frac{\sigma^2 g}{\sigma^2 g + \sigma^2 e}$ , where $\sigma^2 g$ and $\sigma^2 e$ are the variances due to genotype and error, respectively.

Response to selection (Rs) was calculated as $Rs = h_b{}^2 \times i \times \sqrt{\sigma^2 p}$ , where $i$ is the selection intensity at 39% for selected individuals in F6 ($i = 0.984$) and at 92% for selected individuals in F7 and F8 ($i = 0.195$).

The relative traits for each line (%T) were expressed as a percentage of the average of checks at each generation using the following equation: $T\% = \frac{T_L}{T_{Ch}} \times 100$, where $T_L$ is the mean of each line and $T_{Ch}$ is the average of the checks at each generation.

Selection differential and genetic gain were calculated for grain yield and related traits as described by the procedure of St. Martin and McBlain [13], using relative values for the evaluated traits ($T\%$). Calculations of selection differential and genetic gain were performed for the two selection steps available: F6–F7 and F7–F8, according to the following equations: $SD = X_S - X$ and $GG = X'_S - X$, where SD is the selection differential, $X_S$ is the mean of the experimental lines selected from the first stage (F6 or F7) for testing in the second stage (F7 or F8), X is the mean of all experimental lines evaluated in the first stage (F6 or F7), GG is genetic gain, and $X'_S$ is the mean of the experimental lines selected from the first stage and evaluated in the second stage (F7 or F8).

Yield index (YI) was calculated for the selected thirty-four lines and five check cultivars over the three seasons as: $YI = \frac{Y_s}{\overline{Y}_s}$, where $Y_S$ is the grain yield of each genotype and $\overline{Y}_s$ is the average grain yield of check cultivars [14].

Hierarchical cluster analysis was applied to group the evaluated F8 lines according to the level of salt tolerance based on their yield index following the Ward method [16]. Principal component analysis was performed on the averages of the measured traits to determine their relationship. R statistical software version 3.6.1 was used for all statistical analyses.

## 3. Results

### 3.1. ANOVA and Descriptive Evaluation of F6–F8

Analysis of variance for earliness, plant height and yield traits displayed highly significant differences among evaluated genotypes, including advanced lines and check cultivars, in the three generations F6–F8 (Table 1 and Figure 1). The genotypic variance was reduced through successive generations from F6 to F8. Furthermore, a wide range of the evaluated traits was observed among the advanced lines in the three generations (Table 2 and Figure 1). The applied selection intensity was stronger from F6 to F7 (39% of lines promoted to F7) than from F7 to F8 (92% of lines promoted to F8). Therefore, the traits varied largely in F6, followed by F7 and F8. Grain yield varied from 1385 to 4492 kg/ha in F6, whereas it varied from 3518 to 4492 kg/ha in F7 and from 3942 to 4560 kg/ha in F8 (Table 2). In addition, the overall mean of F8 lines was higher than F6 and F7 due to excluding low-yielding breeding lines through selection process in F6 and F7. Moreover, F8 contained advanced lines with higher yield potential than the check cultivars (Table 2).

**Table 1.** Mean squares of studied traits for evaluated genotypes (advanced lines and check cultivars) in the three generations, F6, F7, and F8.

| Generation | Trait | Genotypes | Error | Total |
|---|---|---|---|---|
| | **Df** [†] | **98** | **196** | **296** |
| | Days to heading | 69.24 ** | 2.86 | 24.83 |
| | Plant height | 145.44 ** | 7.58 | 53.23 |
| **F6** | Number of grains/spike | 57.64 ** | 5.55 | 22.76 |
| | 1000-grain weight | 31.59 ** | 3.13 | 12.53 |
| | Grain yield | 2,076,531 ** | 344,898 | 935,619 |
| | Biological yield | 3,948,980 ** | 560,894 | 1,679,154 |
| | **Df** [†] | **41** | **82** | **125** |
| | Days to heading | 46.22 ** | 1.61 | 16.23 |
| | Plant height | 86.66 ** | 4.92 | 31.74 |
| **F7** | Number of grains/spike | 15.15 ** | 1.30 | 5.82 |
| | 1000-grain weight | 9.78 ** | 0.85 | 3.78 |
| | Grain yield | 90,488 ** | 13,672 | 38,665 |
| | Biological yield | 1,234,146 ** | 167,345 | 514,633 |
| | **Df** [†] | **38** | **76** | **116** |
| | Days to heading | 40.77 ** | 1.86 | 14.57 |
| | Plant height | 64.82 ** | 3.22 | 23.47 |
| **F8** | Number of grains/spike | 9.17 ** | 1.09 | 3.75 |
| | 1000-grain weight | 5.75 ** | 0.85 | 2.46 |
| | Grain yield | 39,737 ** | 6120 | 17,095 |
| | Biological yield | 694,176 ** | 89,947 | 286,397 |

[†] Df is degree of freedom, ** indicates *p*-value < 0.01.

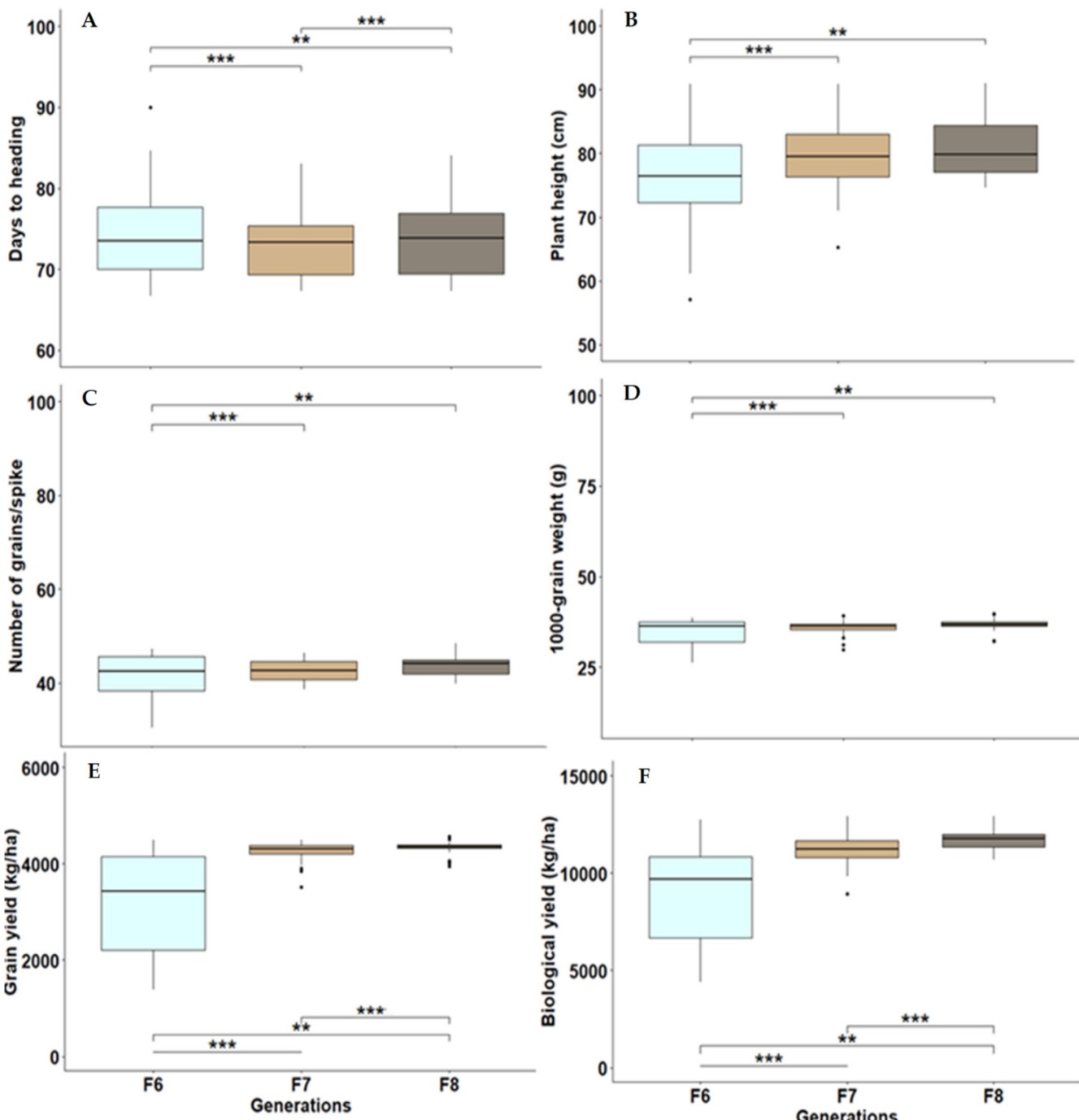

**Figure 1.** Boxplots with median, minimum, maximum and mean values for days to heading (**A**), plant height (**B**), number of grains/spike (**C**), 1000-grain weight (**D**), grain yield (**E**), and biological yield (**F**) for the investigated lines in advanced generations F6, F7 and F8. ** and *** indicate *p*-value < 0.01 and 0.001, respectively.

**Table 2.** Descriptive statistics (minimum, Min; maximum, Max; mean, standard deviation, SD and coefficient of variation, CV) for studied agronomic traits of the evaluated advanced lines and check cultivars in the three generations, F6–F8.

| Generation | Trait | Advanced Lines | | | | | Check Cultivars | | | | |
|---|---|---|---|---|---|---|---|---|---|---|---|
| | | Min | Max | Mean | SD | CV (%) | Min | Max | Mean | SD | CV (%) |
| F6 | Days to heading | 66.67 | 90.00 | 74.09 | 4.78 | 6.45 | 66.33 | 75.67 | 70.20 | 3.44 | 4.90 |
| | Plant height (cm) | 57.10 | 90.90 | 76.34 | 7.03 | 9.21 | 81.67 | 89.33 | 85.87 | 2.97 | 3.46 |
| | Number of grains/spike | 29.97 | 47.29 | 41.49 | 4.61 | 11.11 | 36.94 | 46.06 | 39.98 | 3.58 | 8.95 |
| | 1000-grain weight (g) | 26.30 | 38.70 | 34.91 | 3.27 | 9.36 | 36.79 | 48.93 | 42.64 | 4.47 | 10.48 |
| | Grain yield (kg/ha) | 1385 | 4492 | 3241 | 1015 | 31.30 | 3337 | 4314 | 3906 | 508.0 | 13.01 |
| | Biological yield (kg/ha) | 4400 | 12,736 | 8932 | 1364 | 15.27 | 8872 | 12,368 | 10,843 | 1258 | 11.60 |
| F7 | Days to heading | 67.33 | 83.00 | 73.03 | 3.94 | 5.39 | 65.67 | 73.67 | 69.40 | 2.90 | 4.18 |
| | Plant height (cm) | 65.33 | 90.83 | 79.56 | 5.49 | 6.90 | 75.67 | 87.33 | 81.93 | 4.65 | 5.68 |
| | Number of grains/spike | 38.61 | 46.36 | 42.78 | 2.19 | 5.12 | 37.13 | 45.01 | 41.27 | 3.84 | 9.30 |
| | 1000-grain weight (g) | 29.96 | 39.27 | 36.02 | 1.86 | 5.16 | 38.27 | 45.47 | 41.77 | 3.41 | 8.16 |
| | Grain yield (kg/ha) | 3518 | 4492 | 4254 | 200.9 | 4.72 | 3409 | 4335 | 3951 | 415.0 | 10.50 |
| | Biological yield (kg/ha) | 8950 | 12,916 | 11,224 | 753 | 6.71 | 9693 | 12,453 | 11,204 | 1068 | 9.53 |
| F8 | Days to heading | 67.33 | 84.00 | 73.63 | 3.72 | 5.05 | 66.3 | 74.97 | 70.04 | 3.21 | 4.58 |
| | Plant height (cm) | 74.67 | 91.00 | 80.73 | 4.63 | 5.74 | 82.00 | 90.17 | 87.73 | 3.38 | 3.85 |
| | Number of grains/spike | 39.79 | 48.43 | 43.50 | 2.05 | 4.72 | 36.71 | 45.83 | 42.09 | 3.77 | 8.96 |
| | 1000-grain weight (g) | 32.15 | 39.81 | 36.80 | 1.57 | 4.25 | 37.29 | 49.80 | 41.94 | 4.82 | 11.49 |
| | Grain yield (kg/ha) | 3942 | 4560 | 4327 | 133.1 | 3.08 | 3147 | 4275 | 3824 | 594.2 | 15.54 |
| | Biological yield (kg/ha) | 10,709 | 12,928 | 11,717 | 562.5 | 4.80 | 8635 | 12,374 | 10,578 | 1164 | 11.00 |

### 3.2. Genetic Variability in F6–F8

Phenotypic and genotypic variances ($\sigma^2 p$ and $\sigma^2 g$), phenotypic and genotypic coefficients of variation (PCV, GCV), broad-sense heritability ($h^2_b$), and response to selection (Rs) for the three advanced generations are presented in Table 3. The results reveal that the $\sigma^2 g$ and $\sigma^2 p$, PCV, GCV, and Rs for the measured traits were higher in F6 followed by F7 and F8. The values of GCV were closer to those of PCV for the earliness trait, plant height, 1000-grain weight and number of grain/spike than grain yield and biological yield in the three generations. Moreover, it is worth noting that broad-sense heritability values were high for days to heading, plant height, 1000-grain weight and number of grains/spike while being intermediate for grain yield and biological yield in the three generations under saline conditions (Table 3).

### 3.3. Genetic Gain and Selection Differential

To estimate the progress due to selection, the traits were expressed relatively as a percentage of checks. Obviously, it presents values around 100, thereupon, the values for all traits can be easily followed compared with the average of the checks (Table 4). The selection differential (SD) and genetic gain (GG) were calculated for two selection steps; F6–F7 and F7–F8 (Table 4). The results indicate that the relative traits for F7–F8 step were higher than F6–F7 due to excluding low-yielding lines. As an illustration, the relative grain yield of all F6 lines did not reach checks level (100%), while the average F7 and F8 surpassed the level of the check by 8 and 8.7%, respectively. Consequently, selection step of F6–F7 displayed higher SD and GG than F7–F8.

**Table 3.** Genetic variability parameters for the studied traits in the evaluated bread wheat advanced lines.

| Generation | Parameter [†] | DH * | PH | NG/S | TGW | GY | BY |
|---|---|---|---|---|---|---|---|
| F6 | $\sigma^2 g$ | 20.49 | 43.01 | 16.77 | 8.62 | 688,408 | 1,307,226 |
| | $\sigma^2 p$ | 22.82 | 49.40 | 21.25 | 10.68 | 1,029,216 | 1,860,496 |
| | GCV | 6.11 | 8.59 | 9.87 | 8.41 | 25.60 | 12.80 |
| | PCV | 6.45 | 9.21 | 11.11 | 9.36 | 31.30 | 15.27 |
| | $h^2_b$ | 0.90 | 0.87 | 0.79 | 0.81 | 0.67 | 0.70 |
| | Rs | 4.22 | 6.02 | 3.58 | 2.59 | 667.7 | 943.0 |
| F7 | $\sigma^2 g$ | 14.31 | 26.83 | 3.89 | 2.89 | 27,802 | 403,559 |
| | $\sigma^2 p$ | 15.50 | 30.11 | 4.79 | 3.45 | 40,356 | 567,270 |
| | GCV | 5.18 | 6.51 | 4.61 | 4.72 | 3.92 | 5.66 |
| | PCV | 5.39 | 6.90 | 5.12 | 5.16 | 4.72 | 6.71 |
| | $h^2_b$ | 0.92 | 0.89 | 0.81 | 0.84 | 0.69 | 0.71 |
| | Rs | 0.71 | 0.95 | 0.35 | 0.30 | 26.99 | 104.5 |
| F8 | $\sigma^2 g$ | 12.65 | 19.41 | 3.45 | 2.06 | 12,254 | 228,527 |
| | $\sigma^2 p$ | 13.81 | 21.46 | 4.21 | 2.45 | 17,728 | 316,424 |
| | GCV | 4.83 | 5.46 | 4.27 | 3.90 | 2.56 | 4.08 |
| | PCV | 5.05 | 5.74 | 4.72 | 4.25 | 3.08 | 4.80 |
| | $h^2_b$ | 0.92 | 0.90 | 0.82 | 0.84 | 0.69 | 0.72 |
| | Rs | 0.66 | 0.82 | 0.33 | 0.26 | 17.95 | 79.22 |

[†] GCV is the genotypic coefficient of variation, PCV is the phenotypic coefficient of variation, $h^2_b$ is heritability in broad sense and Rs is response to selection. * DH is days to heading, PH is plant height (cm), NG/S is number of grains/spike, TGW is 1000-grain weight (g), GY is grain yield (kg/ha) and BY is biological yield (kg/ha).

### 3.4. Mean Performance of Promising Lines

The averages of studied traits for the selected thirty-four breeding lines over the three growing seasons are presented in Table 5. The sensitive individuals were excluded through the early generations, and thereby, the selected lines displayed slight variations. Notwithstanding, days to heading significantly differed among the selected breeding lines under salinity stress and ranged between 67.22 and 83.89 days. Lines L72, L78, L59, L24, and L49 displayed the earliest heading, whereas L64, L16, L21, L65, and L23 exhibited the latest heading (Table 5). Plant height considerably varied among genotypes and ranged between 70.98 and 88.76 cm. The tallest plants were assigned to L13, L2, L24, L23, and L27, while the shortest plants were recorded for L53, L57, L72, L34 and L49. Number of grains/spike was significantly divergent among the breeding lines in response to salinity stress, and varied between 39.07 and 47.19 grains. The lines L79, L77, L80, L57, and L21 presented the highest grain number, whereas L53, L45, L54, L78, and L72 displayed the lowest grain number compared with the check cultivars and the other lines under salinity stress. Similarly, 1000-grain weight significantly differed among the breeding lines and ranged between 33.82 and 38.71 g. The heaviest grain index was assigned to L77, L44, L64, L24, and L30, while the lightest grain index was recorded for L45, L16, L53, L27, and L72. Correspondingly, grain yield varied substantially among the breeding lines under salinity stress, ranging between 3962 and 4467 kg/ha. The breeding lines exhibited different performances; L77, L44, L64, L3, and L65 produced the highest grain yield, while L53, L72, L45, L59, and L61 presented the lowest values. In addition, there was a significant contrast regarding biological yield between the lines due to salinity stress, ranging between 9709 and 12,860 kg/ha. The lines L65, L37, L64, L21, and L16 gave the highest biological yield, whereas L53, L59, L45, L54, and L57 presented the lowest values.

**Table 4.** Selection differential (S) and genetic gain (GG) of grain yield and its components for advanced lines in two sets of consecutive generations (F6–F7 and F7–F8), the traits were expressed as the percentage of checks.

| Trait | Generations | First Generation | | Evaluated in Second Generation | S | GG |
|---|---|---|---|---|---|---|
| | | All Lines | Selected Lines | | | |
| Days to heading | F6–F7 | 103.5 | 104.3 | 104.2 | 0.73 | 0.69 |
| | F7–F8 | 104.2 | 104.5 | 104.4 | 0.27 | 0.14 |
| Plant height | F6–F7 | 88.91 | 93.88 | 92.96 | 4.97 | 4.05 |
| | F7–F8 | 92.96 | 94.24 | 93.80 | 1.28 | 0.84 |
| Number of grains/spike | F6–F7 | 102.0 | 105.7 | 104.8 | 3.69 | 2.87 |
| | F7–F8 | 104.8 | 106.3 | 105.2 | 1.46 | 0.33 |
| 1000-grain weight | F6–F7 | 101.2 | 108.2 | 107.5 | 6.92 | 6.28 |
| | F7–F8 | 107.5 | 108.9 | 107.9 | 1.38 | 0.40 |
| Grain yield | F6–F7 | 83.02 | 111.2 | 108.0 | 28.17 | 24.96 |
| | F7–F8 | 108.0 | 109.1 | 108.7 | 1.13 | 0.71 |
| Biological yield | F6–F7 | 80.59 | 103.6 | 100.2 | 23.05 | 19.60 |
| | F7–F8 | 100.2 | 102.4 | 101.9 | 2.23 | 1.70 |

**Table 5.** Mean performance of selected thirty-four breeding lines and five check cultivars over the three generations F6–F8.

| Genotype | Parents | DH * | PH | NG/S | TGW | GY | BY | GY Ranking |
|---|---|---|---|---|---|---|---|---|
| L2 | P1 × P2 | 72.22 | 86.06 | 42.71 | 36.45 | 4314 | 11,201 | 20 |
| L3 | P1 × P2 | 74.67 | 82.68 | 41.29 | 37.53 | 4419 | 11,799 | 4 |
| L13 | P1 × P2 | 76.67 | 88.76 | 41.63 | 36.38 | 4323 | 11,674 | 16 |
| L15 | P5 × P2 | 77.89 | 78.88 | 41.76 | 37.63 | 4367 | 11,811 | 9 |
| L16 | P5 × P2 | 79.33 | 80.41 | 41.28 | 34.66 | 4408 | 11,820 | 6 |
| L21 | P5 × P2 | 78.11 | 77.22 | 45.09 | 37.37 | 4376 | 11,944 | 8 |
| L23 | P5 × P2 | 77.89 | 83.87 | 43.47 | 36.95 | 4363 | 11,320 | 10 |
| L24 | P5 × P2 | 68.67 | 84.06 | 41.76 | 38.05 | 4238 | 10,949 | 29 |
| L27 | P5 × P2 | 77.22 | 82.97 | 43.72 | 35.12 | 4311 | 11,209 | 22 |
| L30 | P9 × P10 | 72.33 | 79.89 | 44.09 | 37.95 | 4355 | 11,479 | 12 |
| L33 | P1 × P3 | 68.89 | 77.23 | 43.60 | 37.09 | 4249 | 11,182 | 26 |
| L34 | P1 × P3 | 75.67 | 72.31 | 44.15 | 37.69 | 4393 | 11,809 | 7 |
| L37 | P1 × P3 | 77.56 | 79.26 | 43.52 | 37.33 | 4340 | 11,967 | 13 |
| L39 | P1 × P3 | 69.56 | 78.11 | 43.88 | 37.04 | 4248 | 11,129 | 27 |
| L44 | P1 × P3 | 72.56 | 78.72 | 43.79 | 38.46 | 4440 | 11,625 | 2 |
| L45 | P1 × P3 | 71.85 | 76.41 | 39.55 | 33.82 | 3987 | 9950 | 37 |
| L49 | P8 × P10 | 68.78 | 74.30 | 43.79 | 37.02 | 4318 | 11,394 | 17 |
| L50 | P8 × P10 | 74.44 | 79.23 | 43.60 | 37.47 | 4302 | 11,209 | 23 |
| L51 | P8 × P10 | 72.11 | 79.44 | 42.39 | 37.16 | 4330 | 11,606 | 15 |
| L53 | P4 × P6 | 69.11 | 70.98 | 39.07 | 34.67 | 3962 | 9709 | 39 |
| L54 | P4 × P6 | 68.89 | 75.97 | 40.12 | 37.58 | 4224 | 10,042 | 30 |
| L57 | P4 × P6 | 72.00 | 71.49 | 45.19 | 36.86 | 4041 | 10,093 | 33 |
| L59 | P4 × P6 | 68.33 | 76.31 | 40.89 | 37.54 | 3989 | 9814 | 36 |
| L61 | P4 × P6 | 69.22 | 74.37 | 40.90 | 37.46 | 4011 | 10,222 | 35 |
| L62 | P4 × P6 | 75.44 | 79.39 | 44.43 | 37.11 | 4340 | 10,978 | 13 |
| L64 | P4 × P6 | 83.89 | 78.57 | 43.81 | 38.21 | 4422 | 11,965 | 3 |
| L65 | P4 × P6 | 78.00 | 76.42 | 42.69 | 37.23 | 4411 | 12,860 | 5 |
| L70 | P1 × P4 | 73.56 | 77.23 | 44.60 | 36.86 | 4316 | 11,586 | 18 |
| L72 | P1 × P4 | 67.22 | 71.76 | 40.68 | 36.00 | 3981 | 10,260 | 38 |
| L77 | P1 × P4 | 73.44 | 78.48 | 45.89 | 38.71 | 4467 | 11,464 | 1 |
| L78 | P1 × P4 | 67.67 | 75.92 | 40.50 | 37.87 | 4038 | 10,175 | 34 |
| L79 | P1 × P4 | 73.67 | 76.99 | 47.19 | 36.32 | 4276 | 11,333 | 25 |
| L80 | P1 × P4 | 69.89 | 78.83 | 45.78 | 37.05 | 4240 | 11,375 | 28 |
| L83 | P10 × P7 | 76.44 | 79.00 | 44.70 | 36.68 | 4314 | 11,015 | 20 |
| Sakha-94 | Check | 73.57 | 82.89 | 43.59 | 38.63 | 4315 | 12,852 | 19 |
| Misr-1 | Check | 74.77 | 81.44 | 41.30 | 38.37 | 4079 | 10,067 | 32 |
| Gemmiza-7 | Check | 72.10 | 86.44 | 40.71 | 37.37 | 4277 | 11,210 | 24 |
| Gemmiza-9 | Check | 73.57 | 88.50 | 44.76 | 37.98 | 4360 | 11,848 | 11 |
| Gemmiza-12 | Check | 70.40 | 86.61 | 45.54 | 36.68 | 4097 | 11,398 | 31 |
| $LSD_{0.05\%}$ | | 1.48 | 2.07 | 1.65 | 1.78 | 102 | 254.2 | |

* DH is days to heading, PH is plant height (cm), NG/S is number of grains/spike, TGW is 1000-grain weight (g), GY is grain yield (kg/ha) and BY is biological yield (kg/ha).

### 3.5. Genotypic Classification according to Salinity Tolerance

The yield index of thirty-four promising lines and five checks was computed and used to classify the evaluated genotypes according to their salinity tolerance. The genotypes were classified into four groups using hierarchical clustering (Figure 2). Group A comprised six genotypes with the highest observed grain yield; hence, they could be considered as highly salt-tolerant genotypes. Group B had seventeen genotypes with high grain yield values; consequently, they are considered as tolerant genotypes. Likewise, group C is composed of seven genotypes that had intermediate grain yield values, and they are considered as moderate salt-tolerant genotypes. Groups D included nine genotypes with lower values of grain yield, and they are considered as slightly salt-tolerant genotypes.

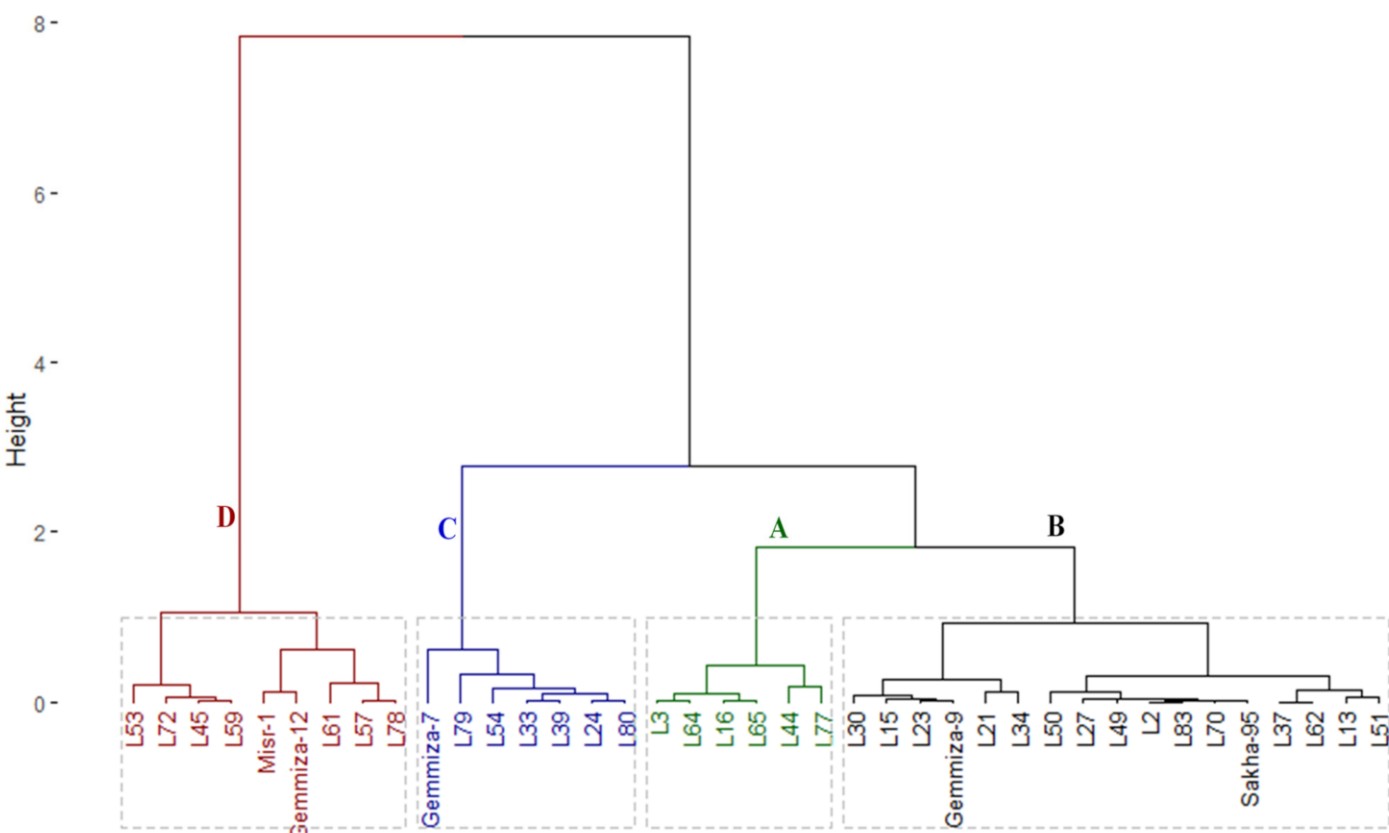

**Figure 2.** Hierarchical clustering of the phenotypic distances among thirty-four F8 wheat lines and five check cultivars based on their yield index.

### 3.6. Interrelationship among Traits

Principal component analysis was applied to visualize the association among the evaluated traits at three advanced generations. The first two principal components displayed most of the variance, reaching about 96.15% (92.51% and 3.64% by PC1 and PC2, respectively); accordingly, they were used to construct the PC-biplot (Figure 3). The trait vectors are represented by acute angles, implying positive associations. Subsequently, a strong positive correlation was detected between grain yield and the number of grains/spike, days to heading, biological yield, 1000-grain weight, and plant height under salinity stress.

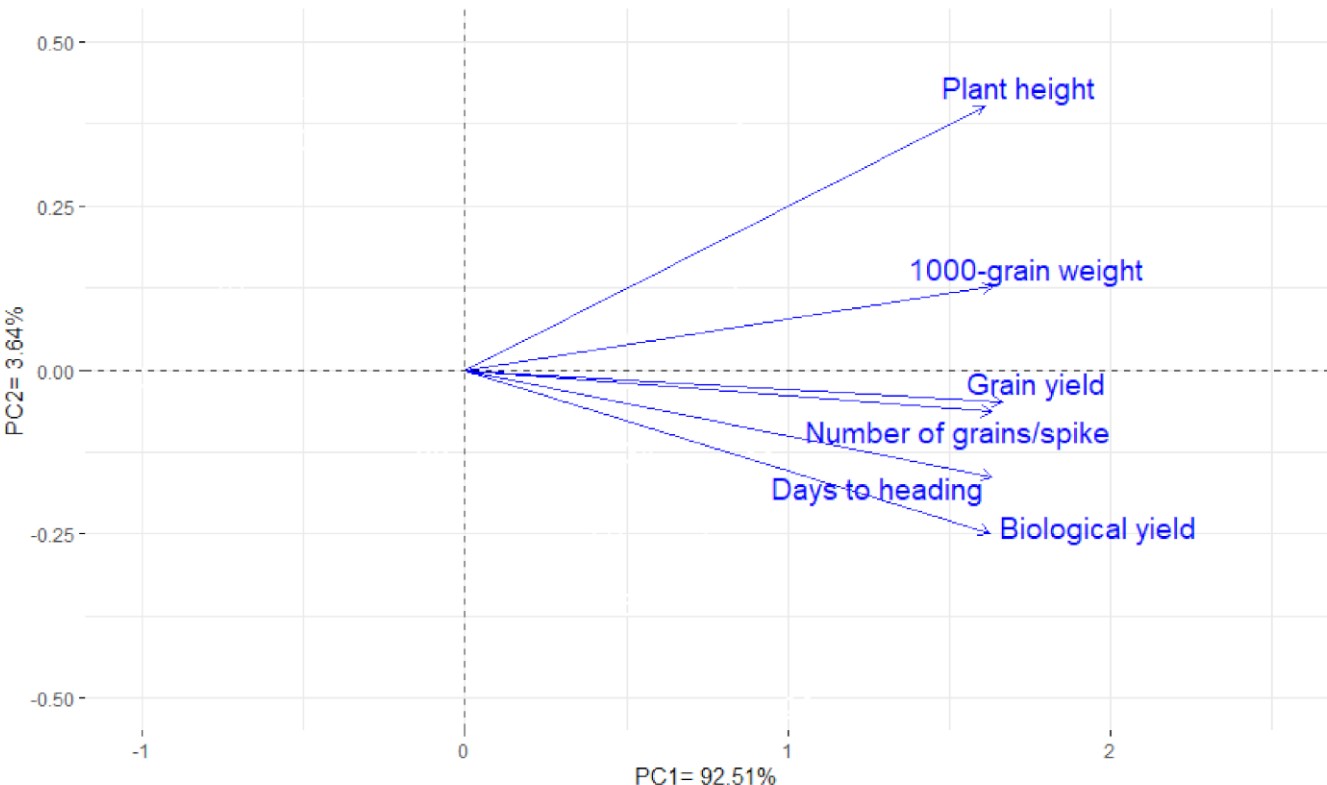

**Figure 3.** Biplot of the first two principal components for agronomic traits of 39 wheat genotypes over three growing seasons under salinity stress.

## 4. Discussion

Salinity in the soil or irrigation water devastatingly impacts wheat growth and productivity. Salinity in the soil can be mitigated by leaching using freshwater or certain agronomic practices to reduce salt from the root zone. Nevertheless, like our study region, in which both soil and irrigation water are salty, the appropriate strategy to cope with these conditions necessitates developing genetically salt-tolerant genotypes. Moreover, developing salt-tolerant wheat genotypes has become more urgent, especially with continuing global population growth and abrupt climate changes [4,8].

Genetic variability is an essential component for developing salt-tolerant wheat genotypes [17]. However, the genetic base of wheat breeding for salt-tolerance is narrow [18,19]. The lack of genetic diversity limits the progress of improving salt-tolerance in wheat. The genetic diversity could be enriched by introgression of new beneficial alleles from exotic genotypes into adopted commercial cultivars [20,21]. This is a vital approach for enhancing current and future breeding efforts of salt-tolerance. In the current study, new breeding lines were derived by crossing exotic genotypes and commercial cultivars. Multiple cycles of selection were implemented for improving yield potential under natural saline field conditions and using saline water in irrigation. Evaluation under actual saline field conditions was effective to exclude salt-sensitive genotypes, particularly in the yielding stage. Similarly, various researchers employed the introgression of favorable alleles from exotic genotypes into adopted wheat cultivars, such as Shavrukov, et al. [22], Akbarpour et al. [18], Al-Ashkar et al. [11], Dadshani et al. [7], Gadimaliyeva et al. [23], and Lethin et al. [24].

The evaluated breeding lines displayed highly significant variations in the three advanced generations, F6–F8, and exhibited a wide range of variability for all studied traits under salinity stress (Tables 1–3, Figure 1). Furthermore, the genotypic variance ($\sigma^2$g) and genotypic coefficient of variation (GCV) demonstrated the presence of inherent genetic differences among the evaluated breeding lines in respect to measured agronomic traits. These genetic divergences of measured agronomic traits reflected the potential

of these lines in enhancing the genetic diversity through favorable alleles that could be exploited through breeding for improving salt-tolerance. Moreover, we noticed a reduction in genotypic and phenotypic variances within the breeding lines in F8 compared with F6 (Tables 1 and 3), demonstrating that the elite lines have become more uniform. Otherwise, the values of phenotypic variance ($\sigma^2 p$) and phenotypic coefficient of variation (PCV) were slightly higher than those of genotypic variance ($\sigma^2 g$) and genotypic coefficients of variation (GCV) for evaluated traits in the three generations. These findings demonstrate the inherent variations among the evaluated lines that remain unaltered by environmental conditions that are beneficial for utilization in breeding for salt-tolerance. In addition, the lower environmental impact on these traits signified the improvement that can be achieved through selection based on phenotypic expression. Additionally, detected intermediate to high broad-sense heritability for evaluated agronomic traits confirmed the efficiency of these traits for direct phenotypic selection to identify salt-tolerant genotypes over the advanced generations. These inferences were strengthened by the detected response of these traits to selection through advanced generations under salinity stress (Table 3). On the other hand, the measured traits displayed a higher response to selection in F6 followed by F7 and F8 [25]. The superior response to selection in F6 compared with F7 and F8 was due to higher phenotypic variation in F6 and also greater selection pressure applied from F6 to F7 that led us to exclude the low potential individuals. Similarly, Venuprasad et al. [26], Bhutta and Hanif [27], Green et al. [28], Akbarpour et al. [18], Okechukwu et al. [29], Oyiga et al. [12], and Lozada et al. [30] elucidated the importance of grain yield and its attributed traits in selection through advanced generations under salinity stress.

Wheat breeders mostly use the pedigree selection scheme to combine complementary characters from different parents to develop new high-yielding genotypes [31]. In the present study, the achieved genetic gain through selection was investigated under salinity conditions based on earliness and yield related traits. The check cultivars are considered as the threshold of agronomic excellence for the program [32]. The overall relative traits of F7 and F8 exhibited significant progress because the breeding lines surpassed the level of the checks compared with F6 (Table 4). Selection through successive inbreeding generations led to accumulating favorable genes in the advanced breeding lines [25]. Accordingly, most of the selected advanced lines exceeded the levels of check cultivars under salinity stress over three generations, particularly; L3, L16, L44, L64, L65, and L77 (Table 5 and Figure 2). Hence, these elite lines can be exploited as newly developed materials with great genetic potential for the further improvement of salt-tolerance. These results emphasize the efficiency of selection to exclude low-potential and salt-sensitive individuals and reaching its objective of producing salt-tolerant genotypes with high-yielding potential. Otherwise, the progress slowed down after F7, as the improvement due to selection was slight compared to F6. The reason for that could be due to lower applied selection intensity from F7–F8 compared to F6–F7 [25]. Consequently, selection differential (SD) and genetic gain (GG) were higher for F6–F7 step than F7–F8 one. In this context, Gracia et al. [25] documented the importance of relative traits expressed as a percentage of the checks, selection differential, and genetic gain in determining the progress that occurred in the Spanish barley breeding program through advanced generations.

Grain yield and its attributed traits are valuable criteria for investigating genotypic responses to salinity stress and identifying salt-tolerant genotypes with high-yielding potential. Number of grains/spike and 1000-grain weight displayed a strong positive association with grain yield, which indicates their importance as vital traits for indirect selection, in particular, in the early generations under salinity stress (Figure 3). These traits are easier to measure compared with grain yield, which is helpful in early generations. A similar positive association between yield and related agronomic traits was proved by El-Hendawy et al. [33], Akbarpour et al. [18], Dadshani et al. [7], Gadimaliyeva et al. [23], and Mansour et al. [8]. Furthermore, a positive association with days to heading reveals grain-filling period requires to be extended to increase grain yield under salinity stress. Moreover, it is worth noting that plant height had a significant positive association with

grain yield and its components under salinity stress. This is a benefit of breeding for salt-tolerance as plant height is considered a surrogate of aerial biomass and is easier to record. Similar findings were depicted by Akbarpour et al. [18], Saade et al. [34], Dadshani et al. [7], and Mansour et al. [8].

## 5. Conclusions

Exotic wheat genotypes were crossed with local cultivars using pedigree scheme under natural salinization conditions to broaden the narrow genetic base of salt-tolerance. Large variability was detected among the evaluated breeding lines for salt tolerance under field conditions at yielding stage in the three evaluated generations, F6–F8. Grain yield and related traits of the selected breeding lines surpassed the level of the adopted check cultivars starting from F7 and F8 under salinity stress. These lines can be exploited as a novel genetic material to enhance breeding for salt-tolerance. These results demonstrate that the selection approach reaches its objective of producing salt-tolerant wheat breeding lines with high-yielding potential. The detected response to selection and high to intermediate broad-sense heritability for studied agronomic traits demonstrated their efficiency in phenotypic selection and identifying salt-tolerant lines over the advanced generations. Number of grains/spike, 1000-grain weight, and plant height displayed a strong positive association with grain yield, which suggests their importance for indirect selection under salinity stress, particularly in the early generations, due to the ease of their measurement.

**Supplementary Materials:** The following are available online at https://www.mdpi.com/2073-4395/11/2/281/s1, Table S1. Name, code, pedigree and origin of the used parents and check cultivars in the breeding program. Table S2. Monthly average minimum temperature (Tmin, °C), maximum temperature (Tmax, °C), growing degree days (GDD, °C) and precipitation (Prec., mm) in the three growing seasons and 38 years' monthly averages (1983–2020). Table S3: Chemical properties of soil and irrigation water at the experimental site

**Author Contributions:** Conceptualization, E.S.A.M., M.M.A.A., M.M.K., M.F.A., A.A.H. and E.M.; methodology, E.S.A.M., M.M.A.A., M.M.K., A.A.H. and E.M.; software, M.M.A.A., M.M.K., M.F.A., A.A.H., and E.M.; validation, E.S.A.M., M.M.A.A., M.M.K., M.F.A., A.A.H., and E.M.; formal analysis, E.S.A.M., M.M.A.A., M.M.K., M.F.A., A.A.H., and E.M.; investigation, E.S.A.M., M.M.A.A., M.M.K., A.A.H., and E.M.; resources, E.S.A.M. and M.F.A.; data curation, E.S.A.M., M.M.A.A., M.M.K., M.F.A., A.A.H., and E.M.; writing—original draft preparation, E.S.A.M., M.M.A.A., M.M.K., M.F.A., A.A.H., and E.M.; writing—review and editing E.S.A.M., M.M.A.A., M.M.K., M.F.A., A.A.H., and E.M.; visualization, E.S.A.M., M.M.A.A. and E.M.; supervision, E.S.A.M., M.M.A.A. and E.M.; project administration E.S.A.M. and M.F.A.; funding acquisition, E.S.A.M., M.M.A.A., M.M.K., M.F.A., A.A.H., and E.M. All authors have read and agreed to the published version of the manuscript.

**Funding:** The current work was funded by Taif University Researchers Supporting Project number (TURSP-2020/111), Taif university, Taif, Saudi Arabia.

**Institutional Review Board Statement:** Not applicable.

**Informed Consent Statement:** Not applicable.

**Data Availability Statement:** The data presented in this study are available upon request from the corresponding author.

**Acknowledgments:** The authors wish to thank Desert Research Center and Zagazig University for the technical and financial support of this research. Appreciation is extended to Ras-Sudr experimental research Station for their help in field activities. The authors extend their appreciation to the Taif University for funding this work through Taif University Researchers Supporting Project number (TURSP -2020/111), Taif University, Taif, Saudi Arabia.

**Conflicts of Interest:** The authors declare no conflict of interest.

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
