# Peer review of "Field Screening of Wheat Advanced Lines for Salinity Tolerance"

_agronomy, doi:10.3390/agronomy11020281_

Round 1

Reviewer 1 Report

The manuscript article by Moustafa et al. entitled 'Field screening of salt-tolerance in bread wheat recombinant inbred lines' is very scientifically sound research work based on the salinity stress tolerant screening of bread wheat RIL populations in three wheat growing seasons.  

The topic is original and novel for its research goal, objectives and used methodologies. The manuscript is written very well and easy to understand language. The conclusions are consistent with the evidence and arguments presented by the authors in this manuscript. They address the main question posed.

I have some following points/queries need to be corrected:

The abstract is too lengthy. Please reduce a bit.

Figure 3: Plant height and TKW are much closer to the yield components, generally it should be just in opposite directions and it is causing very low per cent of PC2, compared to PC1. So, it is advisable to give the proper explanation and the author can try to include some other traits if they have recorded, which can change the PCA graph.

Author Response

Reviewer 1:

The manuscript article by Moustafa et al. entitled “Field screening of salt-tolerance in bread wheat recombinant inbred lines” is very scientifically sound research work based on the salinity stress tolerant screening of bread wheat RIL populations in three wheat growing seasons. The topic is original and novel for its research goal, objectives and used methodologies. The manuscript is written very well and easy to understand language. The conclusions are consistent with the evidence and arguments presented by the authors in this manuscript. They address the main question posed.

Re: We would like to thank the Reviewer for his time dedicated to our manuscript and presenting positive aspects in our manuscript. We highly appreciate the constructive criticisms which improved the manuscript.

I have some following points/queries that need to be corrected:

The abstract is too lengthy. Please reduce a bit.

Re: The abstract has been reduced

Figure 3: Plant height and TKW are much closer to the yield components, generally it should be just in opposite directions and it is causing very low percent of PC2, compared to PC1. So, it is advisable to give the proper explanation and the author can try to include some other traits if they have recorded, which can change the PCA graph.

Re: All measured traits were used in the PC-biplot. In respect of plant height, it displayed a significant positive association with grain yield and its components under salinity stress, which is a benefit to breeding for salt-tolerance as plant height is considered a surrogate of aerial biomass and easier to be recorded. Similar findings were depicted by Akbarpour et al. (2015); Saade et al. (2016); Dadshani et al. (2019) and Mansour et al. (2020). Explanations have been added to the discussion section (lines 410-416)

Reviewer 2 Report

Ehab et al. present an evaluation of genetic variability of new breeding lines in advanced generation under the salt stress condition in wheat. This study is of interest to readers since it will give an idea for developing new salt-tolerant wheat variety in the arid region. Moreover, selection of the pedigree line in the real salt environment will help the variety for better adaptation in saline area. The study and results are presented in detail and the conclusion was well supported. 

I have a few considerations that the authors may consider for improving the manuscript:

  1. Please don’t break the word with the end-of-line hyphenation.
  2. The abstract has 302 words, but according to the journal, the abstract should be a total of about 200 words maximum.
  3. Please keep each equation in the separate paragraph for “2.4 Statistical analysis”.
  4. It’s better to send Table 1 and Table 2 to supplementary and then correct all citations of Tables.
  5. Figure 1 and Table 4 represent the same data. One of them (like Figure 1) should send to supplementary.
  6. What is **** mean in Figure 1? Please mention what is mean by ** and **** in the Figure 1 heading.
  7. I am not clear about Figure 2. How it is connected to over the three generation F6-F8?
  8. Line 70: delete ;
  9. Line 89: “sal-tolerance” -->  “salt-tolerance”
  10. Line 91: F6-8 --> F6-F8
  11. Line 96: “yield-related” --> “yield related”
  12. Line 105: “F7” --> “F7 lines”
  13. Line 110: “3-m” --> “3 m”
  14. Line 120: Table (2) --> Table 2
  15. Line 122: unit is missing after 0-30
  16. Line 208: F6-8 --> F6-F8
  17. Please delete the dot from 1st column 1st row of Table 4: Generation. --> Generation
  18. Please correct 1st column of the Table 4 (it should be F6, F7, F8 )
  19. Line 214: Table (5) --> Table 5
  20. Line 228: Unit is missing for PH and TGW.
  21. Line 237: F7-8 --> F7-F8 and F6-7 --> F6-F7. Also correct these for line 241, 242, 342, 399, 400, 425.
  22. Line 245: Table (7) --> Table 7
  23. Correct 2nd column of Table 6 (it should be like F6-F7 instead of F6-7)
  24. What is “N.” in the 1st column of Table 6? It should be “No.” or “number”
  25. Line 280: Unit is missing for PH and TGW
  26. Please label Group A, Group B, Group C and Group D in Figure 2
  27. Line 393: “improvement salt-tolerance” --> “improvement of salt-tolerance”
  28. Line 424: “yielding-stage” --> “yielding stage”
  29. Please rewrite the last sentence of the Conclusions paragraph.
  30. Line 452: Delete “Please add”

Author Response

Reviewer: 2

Ehab et al. present an evaluation of genetic variability of new breeding lines in advanced generation under the salt stress condition in wheat. This study is of interest to readers since it will give an idea for developing new salt-tolerant wheat variety in the arid region. Moreover, selection of the pedigree line in the real salt environment will help the variety for better adaptation in saline area. The study and results are presented in detail and the conclusion was well supported.

Re: We would like to thank the reviewer for the positive comment regarding our work. We highly appreciate his time dedicated to our manuscript and the constructive criticisms that improved the manuscript.

I have a few considerations that the authors may consider for improving the manuscript:

Please don’t break the word with the end-of-line hyphenation.

Re: The hyphens have been removed from all the text

The abstract has 302 words, but according to the journal, the abstract should be a total of about 200 words maximum.

Re: The abstract has been reduced to be 200 words

Please keep each equation in the separate paragraph for “2.4 Statistical analysis”.

Re: The equations have been arranged in separate paragraphs as the Reviewer suggested

It’s better to send Table 1 and Table 2 to supplementary and then correct all citations of Tables.

Re: Tables 1 and 2 have been moved to supplementary materials

Figure 1 and Table 4 represent the same data. One of them (like Figure 1) should send to supplementary.

Re: After your permission, we would like to keep both of them in the main text. Table 4 (Table 2 in revised version) presents a descriptive evaluation of the advanced lines and check cultivars in the three generations (Min; Max; mean, SD and CV). Figure 1 is helpful to provide a visual summary enabling readers to quickly identify the performance of the advanced lines in the three generations.

What is *** mean in Figure 1? Please mention what is mean by ** and *** in the Figure 1 heading.

Re: Has been added in the heading (line 196)

I am not clear about Figure 2. How it is connected to over the three generations F6-F8?

Re: The heading of Figure 2 has been corrected (line 297).

Line 70: delete ;

Re: the semicolon has been deleted (line 71)

Line 89: “sal-tolerance” -->  “salt-tolerance”

Re: “sal-tolerance” has been changed to  “salt-tolerance” (line 88)

Line 91: F6-8 --> F6-F8

Re: “F6-8” has been changed to “F6-F8” (line 90)

Line 96: “yield-related” --> “yield related”

Re: “yield-related” has been changed to  “yield related” (lines 94, 373)

Line 105: “F7” --> “F7 lines”

Re: “F7” has been changed to  “F7 lines” (line 103)

Line 110: “3-m” --> “3 m”

Re: “3-m” has been changed to  “3 m” (line 107)

Line 120: Table (2) --> Table 2

Re: “Table (2)” has been changed to  “Table S2” (line 116)

Line 122: unit is missing after 0-30

Re: The unit has been added (line 118)

Line 208: F6-8 --> F6-F8

Re: “F6-8” has been changed to “F6-F8” (lines 198, 333 and 424)

Please delete the dot from 1st column 1st row of Table 3: Generation. --> Generation

Re: The dot has been deleted (line 191)

Please correct 1st column of Table 4 (it should be F6, F7, F8 )

Re: The first column has been corrected  (line 198)

Line 214: Table (5) --> Table 5

Re: The parentheses have deleted (line 204)

Line 228: Unit is missing for PH and TGW.

Re: The units have been added (line 218)

Line 237: F7-8 --> F7-F8 and F6-7 --> F6-F7. Also correct these for line 241, 242, 342, 399, 400, 425.

Re: “F7-8” has been changed to “F7-F8” and “F6-7” to “F6-F7” (lines 227, 231, 232, 390 and 392)

Line 245: Table (7) --> Table 7

Re: The parentheses have deleted (line 235)

Correct 2nd column of Table 6 (it should be like F6-F7 instead of F6-7)

Re: The second column has been corrected (line 266)

What is “N.” in the 1st column of Table 6? It should be “No.” or “number”

Re: “N. of grains/spike” has been modified to “Number of grains/spike” (line 266)

Line 280: Unit is missing for PH and TGW

Re: The units have been added (line 270)

Please label Group A, Group B, Group C and Group D in Figure 2

Re: Groups have been labeled in Figure 2 (line 297)  

Line 393: “improvement salt-tolerance” --> “improvement of salt-tolerance”

Re: improvement salt-tolerance” has been changed to “improvement of salt-tolerance” (line 384)

Line 424: “yielding-stage” --> “yielding stage”

Re: “yielding-stage” has been changed to “yielding stage” (line 423)

Please rewrite the last sentence of the Conclusions paragraph.

Re: The sentence has been rephrased (lines 433-437)

Line 452: Delete “Please add”

Re: “Please add” has been deleted (line 460)